# Integrating Oxygen and 3D Cell Culture System: A Simple Tool to Elucidate the Cell Fate Decision of hiPSCs

**DOI:** 10.3390/ijms23137272

**Published:** 2022-06-30

**Authors:** Rubina Rahaman Khadim, Raja Kumar Vadivelu, Tia Utami, Fuad Gandhi Torizal, Masaki Nishikawa, Yasuyuki Sakai

**Affiliations:** 1Department of Bioengineering, Graduate School of Engineering, University of Tokyo, Hongo, Tokyo 113-8654, Japan; tia-utami23@g.ecc.u-tokyo.ac.jp (T.U.); gandhi.rizal@gmail.com (F.G.T.); sakaiyasu@g.ecc.u-tokyo.ac.jp (Y.S.); 2Department of Chemical System Engineering, Graduate School of Engineering, University of Tokyo, Hongo, Tokyo 113-8654, Japan; masaki@chemsys.t.u-tokyo.ac.jp; 3Human Biomimetic System, RIKEN Hakubi Research Team, RIKEN Cluster for Pioneering Research (CPR), Wako 351-0198, Saitama, Japan

**Keywords:** oxygen, permeable, cell fate, early differentiation

## Abstract

Oxygen, as an external environmental factor, plays a role in the early differentiation of human stem cells, such as induced pluripotent stem cells (hiPSCs). However, the effect of oxygen concentration on the early-stage differentiation of hiPSC is not fully understood, especially in 3D aggregate cultures. In this study, we cultivated the 3D aggregation of hiPSCs on oxygen-permeable microwells under different oxygen concentrations ranging from 2.5 to 20% and found that the aggregates became larger, corresponding to the increase in oxygen level. In a low oxygen environment, the glycolytic pathway was more profound, and the differentiation markers of the three germ layers were upregulated, suggesting that the oxygen concentration can function as a regulator of differentiation during the early stage of development. In conclusion, culturing stem cells on oxygen-permeable microwells may serve as a platform to investigate the effect of oxygen concentration on diverse cell fate decisions during development.

## 1. Introduction

The importance of human-induced pluripotent stem cells (hiPSCs) has been explored. Most recently, hiPSCs have helped in in vitro models of human development, and are useful in regenerative medicine, toxicological studies, and disease modeling [1,2]. hiPSCs have self-renewal capacity and can differentiate into any cell type of the human body [3]. Another advantage of using hiPSCs is overcoming ethical limitations in clinical research. An appropriate stem cell niche plays a major role in differentiation [4,5], for instance, in communication to neighboring cells, the extracellular matrix (ECM), physical properties, and environmental characteristics (oxygen concentration) [6]. The effect of oxygen (O_2_) on cell differentiation has been a subject for many years; however, the understanding of the stemness of the cell population is limited [7]. The role of oxygen on cell differentiation can offer insight into reprogramming and committing to a specific cell lineage. Understanding a cell fate decision corresponding to oxygen may help in unlocking the mystery of signalling cascades participating in such phenomena. In the case of in-vivo scenarios, starting from embryogenesis to organogenesis, a cell population or tissue experiences ranges of O_2_ concentrations [8,9]. Therefore, the importance of the effect of oxygen is receiving increasing acknowledgement in the scientific community to recapitulate the in-vivo niche, which is necessary [10].

As mentioned, ranges of oxygen can affect cell differentiation behavior; high oxygen exposure during an early stage of differentiation increases the efficiency of insulin-producing cells [11]. On the other hand, low O_2_ exposure (typically 5% O_2_, 5% CO_2_) facilitates blood vessel formation and maintains normal cardiomyogenesis in early embryonic development [8,12,13,14]. However, it may also be required to shift to a different oxygen concentration setup for the differentiation and maturation of progenitor cells, for instance, by shifting the culture paradigm from a lower-oxygen to an ambient-oxygen environment for the differentiation of hiPSCs into hepatocyte-like cells [15]. Another key event observed during early embryogenesis is symmetrical breaking to establish the cell patterning and polarized expression of germ layer initiates. This morphogenesis is marked by an exit from pluripotency, and an epithelial-to-mesenchymal transition in preparation of entering the differentiation state [16,17]. Such models should be available with high throughput for screening the role of the microniche in generating 3D cultures.

Differentiation into derivatives of all three germ layers can be employed as a platform for specific disorder modeling that is specifically hypoxic-driven [18]. In this regard, a hypoxic environment assists in mesodermal expression [19]. Low O_2_ can enhance the differentiation efficiency of human PSCs into hepatic [20], neuronal [21,22], cardiac [23], endothelial [24], and chondrogenic [25] cell types. In another study, preconditioned PSCs at low O_2_ affected the cell commitment to early germ layer formation during successive ambient oxygen culture conditions, specifically increasing the gene expression of mesodermal and endodermal lineages [26]. Another study also revealed the upregulation of endodermal genes in a low O_2_ environment as compared to a high O_2_ one [27]. Numerous protocols for the differentiation of 3D and 2D monolayer cultures of hiPSCs were established to specific lineages [28,29,30,31], especially focusing on the application of growth factors. However, optimal culture conditions considering environmental cues responsible for differentiation in in vitro are progressing. Moreover, the effect of O_2_ on differentiation in 3D using hiPSCs is still not completely understood. In our culture platform, we used oxygen-permeable microwells to allow cell aggregation in 3D. As an oxygen-permeable sheet, we used polydimethylsiloxane (PDMS) for the fabrication of honeycomb-shaped microwells.

In this study, we sought to assess the differentiated state of hiPSCs under the influence of oxygen ranging from a hypoxic to an ambient O_2_ culture environment. For the generation of 3D aggregates, we used a simple platform, i.e., honeycomb-shaped microwells, as previously described by [32,33,34]. The growth of the aggregates was monitored over the span of the culture, and the expression profiles of three germ layers were analyzed, strongly maintaining pluripotency. Understanding the effect of oxygen on differentiation could lead to reproducing the complexities of the in-vivo niche for modeling morphogenetic events.

## 2. Result

### 2.1. Self-Assembled Aggregate Formation on Microwells

The hiPSCs deposited on the honeycomb-shaped microwells self-assembled to form aggregates, as illustrated in Figure 1A,B. We considered cell densities of 1.56 × 10^4^, 6.25 × 10^4^, and 12.5 × 10^4^ cells/cm^2^ in order to predetermine the optimal cell density for further analysis. The cell densities did not show high influence on the aggregation of cells, and the morphology of budding was maintained in all conditions, as shown in Figure 1C. We chose 6.25 × 10^4^ cells/cm^2^ to carry out other experiments. The formation of the bud can be observed in Figure 2A, starting from Day 0 to Day 5. They started to aggregate within 24 h, forming spheres like 3D constructs. The budlike morphology was observed by Day 5 of the culture system. The formation of cell aggregates is unique in terms of the occurrence of the budlike morphology, which may have been related to the culture platform in which we allowed the aggregation on PDMS-based microwells. On the other hand, the morphology of the aggregates did not resemble budlike growth in the U-shaped bottom and TCP (Tissue culture plastic) plate as depicted in Appendix A. The microwells enabled the homogeneous distribution of cells, avoiding the formation of larger aggregates (>300 µm).

The aspect ratio was measured as the ratio of the major axis (L1) to the minor axis (L2), as depicted in Figure 2B. When these 3D constructs were exposed to different oxygen environments ranging between 2.5 and 20%, the aspect ratio of Day 2 was approximately almost 1, indicating a spherical morphology. However, on Day 5, due to the budlike extended growth, they varied significantly (Figure 2C). On the basis of the evaluation of the average size of the aggregates from Figure 2D, the average size of aggregates compared between different oxygen levels on Day 5 was also significantly higher, corresponding to a higher level of oxygen (Figure 2C). Additionally, the average size of the aggregate was higher in the cases of 10% and 20% O_2_. This may have been due to the availability of oxygen for their growth.

### 2.2. Cellular Metabolism of hiPSCs Aggregates

On the basis of the live dead staining of the aggregates using calcein AM and propidium iodide of the aggregates on 5 days of culture, no significant differences in cell viability were observed (Figure 3A(i–iv)). However, there were dead cells stained red, as shown by white arrows, scattered and not found in the construct. These dead cells were probably not cohesively attached to the aggregates. To add value to the qualitative analysis of cell viability, we performed semi-quantitative analysis using a Cell Titer Glow^®^ Luminescent cell viability assay, which is shown in Figure 3B. This method of cell viability quantification relies on the presence of ATP, signalling the presence of metabolically active cells. There was a significant difference between the 2.5% and 20% O_2_ groups. However, the luminescence was similar when comparing the 5%, 10%, and 20% O_2_ groups. Higher luminescence was observed in the 20% O_2_ group, which could also have been related with the larger size of the aggregates, as observed in Figure 2D, suggesting the representation of cell proliferation (larger aggregates) as influenced by oxygen. Accordingly, we measured the extracellular glucose concentration to assess the availability of glucose for its sustenance in the hypoxic environment.

The glucose consumption concentration was enough to metabolize for energy, even at Day 5 of culture, as shown in Figure 3C, irrespective of oxygen concentration. Glucose consumption and lactate production at Day 5 were low in the case of 5% O_2_ (glucose consumption, 7.55 mM; lactate production, 8.89 mM) when compared to other conditions. Moreover, the lactate concentration collected from the culture medium increased from Day 3 to Day 5 (Figure 3D). Interestingly, the lactate concentration in 5% O_2_ was lower than that in other conditions. However, the ratio of lactate production to glucose consumption depicts a more accurate metabolic assessment. The production of lactate by the aggregates utilizing the glucose was more rapid in 5% O_2_ (1.309) and 2.5% O_2_ (1.21), higher than that in 10% O_2_ (1.04) and 20% O_2_ (1.123), indicating that 5% and 2.5% O_2_ relied more on the anaerobic metabolic pathway for obtaining energy, as shown in Figure 3E, more significantly at Day 5 in the cases of 5% and 2.5% O_2_. The anaerobic metabolism under a hypoxic environment plays a role in the maintenance of pluripotency [35].

### 2.3. Exit from Pluripotency

The exit from pluripotency was assessed through the segregated expression of OCT4 (pluripotent) and OTX2 (ectoderm) (Figure 4A(i,ii)). Upon evaluation of gene expression of pluripotent markers in different oxygen setups (Figure 4B), the expression of NANOG was upregulated in 5% O_2_ as compared to other oxygen levels. The high expression of NANOG could be correlated to the anaerobic metabolism in hypoxic conditions [36,37]. However, the expression of SOX2 and OCT4 was maintained in all conditions.

Furthermore, the epithelial–mesenchymal transition was observed by staining with E-cadherin and N-cadherin (Figure 4C), and the gene expression of E-cadherin and N-cadherin in different oxygen levels aligned with 5% O_2_ and 10% O_2_, respectively (Figure 4D). The expression of pluripotent markers was on par with the expression of E-cadherin in the case of 5% O_2_ and vice versa. We hypothesized that the pluripotency was maintained while the segregated region entered the differentiated state, which was specifically higher in the case of 5% O_2_. The shift from epithelial to mesenchymal cells is crucial in cell fate decisions and aids in the dynamics of cell differentiation [35].

### 2.4. Oxygen Affecting Three Lineage Gene Expression

The colocalized expression of differentiation signature was observed. To elucidate the correlation between oxygen and differentiation on the aggregates, we performed immunocytochemistry using three lineage markers representative of each group, namely, OTX2 (ectoderm), Brachyury (mesoderm), and GATA4 (endoderm) (Figure 5A). The semiquantitative measurement of fluorescence intensity over DAPI exhibited the upregulation of OTX2 and GATA4 at 5% and 10% O_2_. However, regarding Brachyury, the expression was maintained between 5% to 20% O_2_. The result corresponds to the gene expression of three germ lineages analyzed with qRT-PCR (Figure 5B–D). In general, ectodermal markers OTX2 and Nestin upregulated significantly higher in 5% and 10% O_2_ as compared with other extreme ends of the oxygen spectrum. However, PAX6 was maintained in all the conditions. Mesodermal markers Brachyury, RUNX1, and TBX6 were significantly higher in 5% O_2_ as compared to other oxygen levels.

In addition, endodermal markers such as GATA4 were upregulated in 5% O_2_, and GATA6 was maintained in all the conditions. Interestingly, FOXA2 was significantly higher in 5% O_2_ compared to 2.5%, 10%, and 20% O_2_. This could have been due to endoderm enrichment under hypoxic conditions [20].

In the gene expression of three germ layers, aggregates grown in 5% O_2_ showed a relatively higher expression of GATA6 (endoderm) and Nestin (ectoderm), which may exhibit certain aspects of neuroendodermal rudiments, as shown in Figure 5E,F. GATA6 is responsible for mediating the exit from pluripotency and is a central regulator for the development of the human definitive endoderm. On the other hand, Sox2 plays several roles in developmental processes, such as the maintenance of undifferentiated iPSC, is a neural regulator, and participates in gut endoderm patterning. Here, we observed the coexpression of GATA6 and Sox2 on the pluripotent region, and the segregated expression of brachyury on the differentiated region may indicate the emergence of a mesendoderm (Brachyury and GATA6), and further development towards the formation of endodermal derivatives [38,39]. Additionally, neural markers N-CAD and Nestin were mainly observed in the differentiated region of the aggregates, and showed certain aspects of neuroendodermal rudiments [40].

## 3. Discussion

In-vitro modeling using hiPSCs as a cell source for differentiation has been unlocking key aspects in tissue engineering and regeneration. Studying the cell response to the microniche of stem cells has led to a deeper understanding of how to design and tune the cellular microenvironment in order to achieve the goal of tissue engineering. The switch from 2D to 3D in vitro models has been a continuous effort in the research community. In our model, we aimed to investigate the role of oxygen as a single factor for influencing the early differentiation. A hypoxic environment for preconditioning the growth of pluripotent stem cells was shown to be essential for directing differentiation [26] in a 2D hiPSC model. In our model, the aggregates formed in PDMS-based microwells with oxygen permeation allowed for the self-assembled 3D construct with a bud-like morphology, showing the expression of differentiated markers in the bud region, and pluripotent markers in the main body. Our results suggest that a low-oxygen environment leads to comparatively small aggregates and decreased growth. The glycolytic pathway relies on energy utilization in a low-oxygen environment. These findings seem to enhance the early differentiation of the aggregates in a hypoxic condition. For example, at 5% and 10% O_2,_ there was upregulation of FOXA2 (endoderm) [20]. Ectodermal and mesodermal markers also showed similar responses with the upregulation of OTX2, Nestin, Brachyury, RUNX1, and TBX6. In addition, the upregulation of OTX2 and GATA4 found in gene expression and colocalized expression from immunocytochemistry may show certain characteristics of neuroendodermal rudiments. Interestingly, the bud-like morphology that was observed may exhibit certain aspects of symmetrical breaking, leading to the self-organization and morphogenesis that was shown by the expression of OTX2 and OCT4, and EMT changes [16,17].

Even in the abundance of oxygen, they switch to the glycolytic pathway in order to meet the high demands of energy, which we also observed in the case of 20% O_2_. The metabolic signatures between hiPSC and hESC are different; however, hiPSC mostly relies on glycolysis [41]. This study also showed the correlation between the cellular metabolism and the maintenance of pluripotency. The lactate production rate was higher, indicating the anaerobic metabolism to meet the energy requirements, in turn upregulating NANOG expression, mainly in 5% O_2_ [35]. E-cad/N-cad confirmed the exit from pluripotency. The high gene expression of the ectoderm, mesoderm, and endoderm in the bud region alongside the pluripotent markers on the main body suggests that pluripotency was maintained. 

It would be interesting to incorporate the exogenous growth factors in addition to oxygen preconditioning, and careful timing can aid in early differentiation before proceeding to a later stage of differentiation, for example, via preconditioning in a hypoxic environment for the generation of lung progenitor cells at an early stage [26] and the enhanced expression of endodermal markers [20]. In summary, the aggregates grown on PDMS with different oxygen level setups influenced the differentiation capacity while maintaining pluripotency. The 5% and 10% O_2_ levels can be considered for showing improved early differentiation markers. In the future, the current platform could be further advanced by incorporating other regulating factors with the methodical timing of oxygen exposure at the initial stage of differentiation, and it could improve the organoid-based culture that follows directed differentiation. An alternative approach of forming aggregates on permeable microwells with initial low oxygen exposure can help in understanding the early lineage commitment during developmental events [8,42,43], and has potential to construct in-vitro models.

## 4. Materials and Methods

### 4.1. Fabrication of Honeycomb Microwells and Sterilization

A silicon mold was used to fabricate polydimethylsiloxane (PDMS) honeycomb microwell sheets, which was prepared with SU-8-based negative photolithography (MicroChem, Newton, MA, USA). The surface of the mold was coated with CHF_3_ to help in peeling off PDMS sheets, as previously described by [32,33,34,44]. The patterned SU-8 acted as a replica mold for PDMS. Honeycomb-based PDMS microwell sheets (diameter: 326 µm and depth: 360 µm) were generated using replica mold, punched into a 6.5 mm diameter size using a biopsy punch, and inserted into PDMS bottom 96-well plates [32,33,34,45,46]. The sheets were coated with 1% (*w*/*v*) Pluronic F-127 (P2443 Sigma-Aldrich, St. Louis, MO, USA) and left overnight at 4 °C to ensure cell aggregation without any cell attachment. The sheets were UV-sterilized for 30 min and washed with phosphate buffer saline (PBS) prior to using them for further experiments.

### 4.2. Monolayer hiPSC Culture

hiPSC cell line TKDN-4M was obtained from the Stem Cell Bank Centre for Stem Cell Biology and Regenerative Medicine, University of Tokyo (Tokyo, Japan) [47]. It was maintained in a vitronectin (Thermo Fischer Scientific, Waltham, MA, USA) coated tissue culture dish (Iwaki, Japan) in complete supplemented Essential 8 (E8) culture medium (Thermo Fischer Scientific, Waltham, MA, USA) following the manufacturer’s recommendations with daily culture medium changes, and a passage number < 40 was used for experiments.

### 4.3. Aggregate Formation on Honeycomb Microwells

The hiPSC monolayer was treated using 0.5 mM ethylenediaminetetraacetic acid (EDTA) in a PBS solution for 5–10 min, followed by gentle dissociation with E8 medium sprayed to collect the cells. The cell suspension was filtered through a 40 µm cell strainer (Corning) to have a single-cell suspension, and counted before seeding on the microwell sheet fitted in 96-well plates with a complete E8 culture medium supplemented with 10 µM Y27632 (Wako, Japan). The optimal cell density was determined by inoculating 1.56 × 10^4^ cells/cm^2^ (5000 cells/well), 6.25 × 10^4^ cells/cm^2^ (20,000 cells/well) and 12.5 × 10^4^ cells/cm^2^ (40,000 cells/well). The optimal cell density was determined and used for further experiments. The oxygen concentration was varied by maintaining the culture in 37 °C multigas incubators with different oxygen levels, namely, 2.5%, 5%, 10% and 20%. The aggregate formation was observed using an inverted microscope (Zeiss Axiovert A1 K26-12). The medium was replaced every 2 days. In order to show the formation of aggregates in different conditions, we had a U-shaped bottom plate and 96-well TCP plate as the control, as shown in Appendix A.

### 4.4. Morphological Analysis

In order to characterize the aggregates, the images were captured using an inverted microscope, and analyzed using ImageJ software (National Institutes of Health, Bethesda, MD, USA). The aspect ratio was measured as shown in Figure 2B, and the size of the aggregates was also measured. Additionally, live dead staining was performed to evaluate the cell viability corresponding to aggregates grown at different oxygen levels. The aggregates were stained with a 1:1000 dilution of both calcein AM (excitation: 490 nm, emission: 515 nm) for green fluorescence, and propidium iodide (PI) (excitation: 535 nm, emission: 617 nm) for red fluorescence. Viable cells appeared to be green, whereas dead cells appeared to be red. Additionally, a Cell Titer Glow^®^ Luminescent Cell Viability Assay was performed to quantify the number of viable cells. The amount of ATP was quantified by cell lysis indicating the number of viable cells. The Cell Titer Glow^®^ reagent was added in equal volume as that of the culture medium, and incubated for 10 min at room temperature before measuring the luminescent signal using a multiplate reader (Wallac Arvo™ 5X USA).

### 4.5. Glucose and Lactate Measurement

To assess the cellular metabolism, the culture supernatants collected on Days 3 and 5 were analyzed using a BD-7D bioanalyzer (Oji Scientific, Japan). The amount of extracellular glucose and lactate was quantified. Glucose consumption was measured from the relative amount of quantified glucose concentration collected from the supernatant to the base glucose concentration of the fresh medium. The lactate production rate was the ratio of the amount of lactate production to the amount of consumed glucose.

### 4.6. Quantitative Real-Time PCR

The mRNA was isolated from the aggregates using a TRIzol reagent (Thermo Fischer Scientific, Waltham, MA, USA), followed by cDNA synthesis using ReverTra Ace qPCR RT Master Mix (Toyobo, Japan). The relative gene expression was performed using Thunderbird SYBR qPCR mix (Toyobo, Japan) as per the manufacturer’s instructions, and the samples were run on a StepOnePlus (Applied Biosystems, Waltham, MA, USA) system. The primer sequences are listed in Table 1.

### 4.7. Immunocytochemistry

After 5 days of culture, the hiPSCs aggregates were collected and fixed with 4% paraformaldehyde overnight at 4 °C. The aggregates were treated with 1% Triton X-100 dissolved in PBS for 1 h at room temperature, followed by 1× PBS wash three times. Further, it was incubated in a blocking buffer for 2 h, followed by three PBS washes prior to depositing the primary antibodies. The incubation with primary antibodies was carried out for 12 h; the antibodies were diluted in blocking buffer with 0.2% (*v*/*v*) Tween 20 to allow for optimal antibody infiltration to the 3D aggregates. The appropriate concentrations of primary antibodies used were: anti-OCT4 (mouse, 15 µg/mL, SC-5279 Santa Cruz Biotechnology), anti-OTX2 (goat, 15 µg/mL, AF1979 R&D Systems), anti-OTX2 (mouse, 15 µg/mL, MAB1979 R&D Systems), anti-Brachyury (rabbit, 15 µg/mL, MAB20851 R&D Systems), anti-GATA4 (goat, 15 µg/mL, SC-1237 Santa Cruz Biotechnology), anti-E-cadherin (mouse, 10 µg/mL, MAB1838 R&D Systems), anti-N-cadherin (sheep, 10 µg/mL, AF6426 R&D Systems), anti-GATA6 (mouse, 10 µg/mL, MAB1700 R&D Systems), anti-Sox2 (mouse, 10 µg/ml, MAB2018 R&D Systems) and anti-Nestin (mouse, 10 µg/mL, MAB1259, R&D Systems). After repeated washing, secondary antibodies (dilution 1:500) conjugated with Alexa fluorochromes were added to the samples for 3 h in the dark at room temperature. The secondary antibodies used were donkey anti-mouse AlexaFluor 488 (ab150105, Abcam, Cambridge, United Kingdom), donkey anti-goat AlexaFluor 555 (ab150130, Abcam), donkey anti-mouse AlexaFluor 647 (ab150107, Abcam), donkey anti-rabbit AlexaFluor 488 (ab150073, Abcam) and donkey anti-sheep AlexaFluor 488 (ab150177, Abcam). Repeated washing with PBS was performed, followed by counterstaining with 1:1000 diluted DAPI (Donjindo, Japan) for 1 h at room temperature, washed three times with PBS, and samples were imaged under an FV3000 confocal microscope (Olympus, Japan). The merged Z-stacked images were used as representative images from each group.

### 4.8. Statistical Analysis

Data are represented as the mean ± SEM obtained from at least three independent experiments. Statistical analysis was performed using one-way ANOVA with Tukey’s multiple comparisons in GraphPad Prism software (v.9.3.1), with *p* < 0.05 considered to be significant.

## Figures and Tables

**Figure 1 ijms-23-07272-f001:**
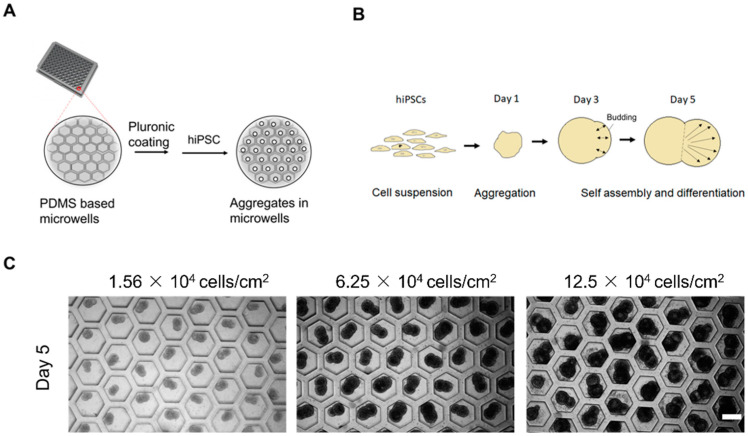
Experimental setup, cell seeding, and mechanism. (**A**) Schematic concept of the preparation of microwells prior to cell seeding. (**B**) Schematic illustration of the self-assembled aggregates. (**C**) Aggregates after Day 5 versus cell seeding density, scale bar: 200 µm, N = 3 sets of independent experiments.

**Figure 2 ijms-23-07272-f002:**
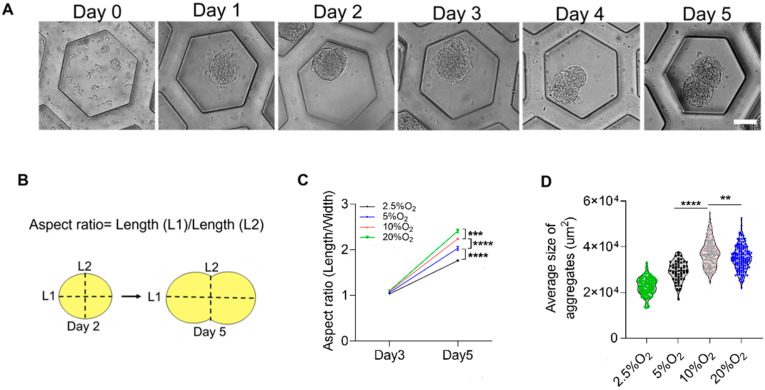
Formation and characterization of the aggregates. (**A**) Aggregates forming through 5 days of culture in honeycomb-shaped microwells, scale bar: 50 µm. (**B**) Schematic illustration of measuring the aspect ratio and comparing Days 2 and 5. (**C**) Aspect ratio of the aggregates at Days 2 and 5. (**D**) The average size of the aggregates versus oxygen. Statistical analysis was performed with one-way ANOVA/Tukey’s test. ** *p* < 0.01, *** *p* < 0.001, **** *p* < 0.0001; ns = non-significant with N = 3 sets of independent experiments.

**Figure 3 ijms-23-07272-f003:**
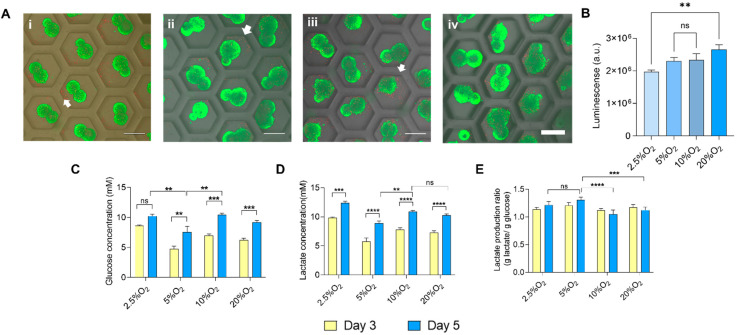
Cell viability and metabolism of the hiPSC aggregates. (**A**) Live dead staining of aggregates grown in (**i**) 2.5%, (**ii**) 5%, (**iii**) 10%, and (**iv**) 20% O_2_, scale bar: 200 µm. (**B**) Quantitative analysis of cell viability using CellTiter Glo^®^ assay at Day 5. (**C**) Glucose consumption of the aggregates at Days 3 and 5. (**D**) Lactate was secreted in the culture supernatant at Days 3 and 5. (**E**) Lactate production ratio depicting lactate secretion to that of glucose consumption at Days 3 and 5. Statistical analysis was performed with one-way ANOVA/Tukey’s test. ** *p* < 0.01, *** *p* < 0.001, **** *p* < 0.0001; ns = non-significant with N = 3 sets of independent experiments.

**Figure 4 ijms-23-07272-f004:**
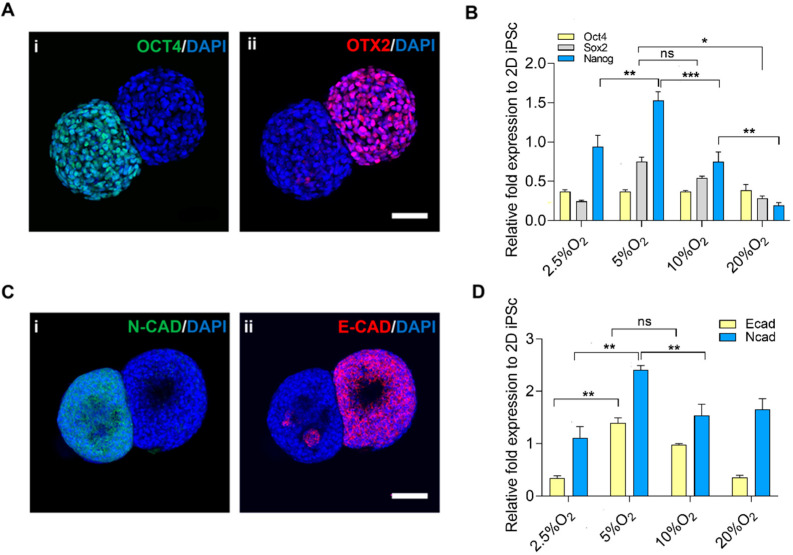
Exit from pluripotency and epithelial–mesenchymal transition. (**A**) Localized expression of the pluripotent marker by the presence of (**i**) OCT4 and (**ii**) OTX2, scale bar: 50 µm. (**B**) Gene expression of pluripotent markers, corresponding to oxygen. (**C**) Exit from pluripotency and EMT, (**i**) E-CAD (**ii**) N-CAD, scale bar: 50 µm. (**D**) Gene expression of EMT. Statistical analysis was performed with one-way ANOVA/Tukey’s test. * *p* < 0.05, ** *p* < 0.01, *** *p* < 0.001; ns = non-significant with N = 3 sets of independent experiments.

**Figure 5 ijms-23-07272-f005:**
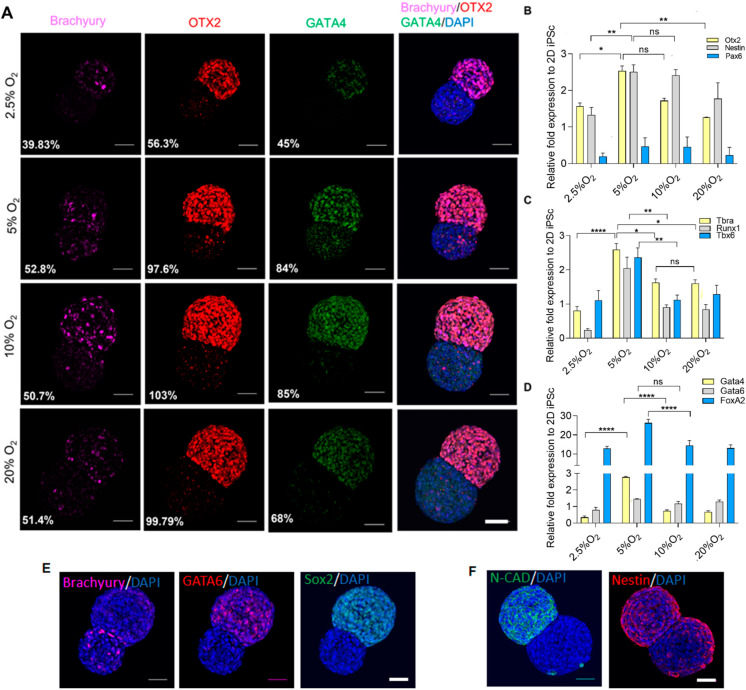
Gene expression of three lineages versus oxygen. (**A**) Comparison of three germ expression markers by immunocytochemistry between 2.5% to 20% O_2_, scale bar 50 µm. (**B**) Gene expression of ectodermal, (**C**) mesodermal, and (**D**) endodermal markers. (**E**) Aggregates grown at 5% O_2_ from Day 5 showing co-expression of GATA6 and Sox2, and segregated Brachyury; (**F**) Expression of N-CAD and Nestin. Scale bar, 50 µm. Statistical analysis was performed with one-way ANOVA/Tukey’s test. * *p* < 0.05, ** *p* < 0.01, **** *p* < 0.0001; ns = non-significant with N = 3 sets of independent experiments.

**Table 1 ijms-23-07272-t001:** Primer sequences used for qRT-PCR analysis.

Gene	Forward Primer	Reverse Primer
OCT4	AGTGGGTGGAGGAAGCTGACAAC	TCGGTTGTGCATAGTCGCTGCTTGA
SOX2	GGCAGCTACAGCATGATGCAGGAGC	CTGGTCATGGAGTTGTACGCAGG
Nanog	AGGACAGGTTTCAGAAGCAGAAGT	TCAGACCATTGCTAGTCTTCAACC
E-cadherin	AGCCCTTACTGCCCCCAGAG	GGGAAGATACCGGGGGACAC
N-cadherin	CAACGGGGACTGCACAGATG	TGTTTGGCCTGGCGTTCTTT
OTX2	GGAGAGGACGACATTTACTAGG	TTCTGACCTCCATTCTGCTG
Nestin	GCGTTGGAACAGAGGTTGGA	TGGGAGCAAAGATCCAAGAC
PAX6	GAGTGCCCGTCCATCTTTG	GTCTGCGCCCATCTGTTGCTTTTC
Brachyury	ATCGTGGACAGCCAGTACG	GCCAACTGCATCATCTCCAC
RUNX1	CCCTAGGGGATGTTCCAGAT	TGAAGCTTTTCCCTCTTCCA
TBX6	AAGTACCAACCCCGCATACA	TAGGCTGTCACGGAGATGAA
GATA4	AGCACACTGCATCTCTCCTGTG	CTCCGCTTGTTCTCAGATCCTC
GATA6	CCCACAACACAACCTACAGC	GCGAGACTGACGCCTATGTA
FOXA2	GCATTCCCAATCTTGACACGGTGA	GCCCTTGCAGCCAGAATACACATT

## Data Availability

The data presented in this study are available in the article and Appendix A. The raw data are available on request from the corresponding author.

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
