# Peer review of "Integrating Oxygen and 3D Cell Culture System: A Simple Tool to Elucidate the Cell Fate Decision of hiPSCs"

_ijms, 2022, doi:10.3390/ijms23137272_

Round 1

Reviewer 1 Report

The manuscript entitled “Integrating oxygen and 3D cell culture system: a simple tool to elucidate cell fate decision of hiPSC” authored by Khadim et al. demonstrated that different oxygen levels combined with 3D-like culture on PDMS micro-wells could influence the pluripotency and differentiation capacity of human iPSCs. Particularly, the importance of oxygen as prime factor for localized expression of pluripotent and differentiation markers was highlighted. The study was technically sound, and the presentation was clear to read.

  1. Could authors mention how they maintained O2 levels? Any hypoxia chamber used? Include the details in the manuscript.

Author Response

Dear Reviewer,

We would like to thank the reviewer for careful revision of our manuscript. Please find the attachment.

Reviewer 2 Report

Dear Authors,

In this manuscript, you cultured 3D aggregates made from hiPSCs in oxygen permeable microwells, evaluating multiple parameters associated to aggregate formation, cell pluripotency and three-lineage formation. 

You aimed to assess the effect of oxygen on the differentiation state of hiPSCs.

I have some comments on the current version of the manuscript:

* Figure 2: Higher oxygen concentrations lead to significantly larger aggregates, appearing as significant gain. This does not lead to a higher glucose consumption rate (Figure 3), which also seems an advantage to me.
* Figure 2: were the cells spinned to induce the formation of the aggregate or was that made by self-assembly and reorganization of cells within the honeycomb well?
* Figure 3: what was the rationale of a live-dead stainig if this figure lack of a proper quantification? Was this method able to pick-up dead cells in the putative necrotic core of larger aggregates?
* Figure 3: How often was the medium refreshed to assess glucose consumption and lactate production? You assess on line 140-141 that the "lactate concentration in 5% O2 was lower than other conditions", but then the panel in Figure 3D does not seem to be coherent with 3B and 3C. At 5% CO2, the aggregates have a higher concentration of glucose than lactate, but in Panel 3D the ratio is between lactate and glucose. Could you please clarify a bit better this section?
* Lines 241-242: it is not clear to me what "both of which were balanced" means in this context.
* Differentiation potential was assessed by checking at different lineage markers. However, no information was provided on the actual differentiation potential, by generating cells from the multiple layers or even specified derivatives. Are oxygen concentrations affecting the differentiation potential towards neurons, intestine, cardiomyocytes, etc? Examples with different cell types would have significantly enriched the manuscript.
* What conclusions can we draw from this study? Should we start culture iPSCs in low oxygen concentation, due to the apparent enhanced expression of differentiation markers? Should we keep culture them in high glucose concentration due to the higher growth rate and size of the aggregates? Please provide a "take-home message" from this small set of experiments. 

Author Response

Dear Reviewer,

We would like to thank the reviewer for careful reviewing our manuscript and appreciate the constructive comments. Please find the attachment.

Round 2

Reviewer 2 Report

Dear Authors,

In this manuscript, you cultured 3D aggregates made from hiPSCs in oxygen permeable microwells, evaluating multiple parameters associated to aggregate formation, cell pluripotency and three-lineage formation. 

You have made some minor changes to the previous version, which overall improved the manuscript, but I still consider the results of this manusript quite limited.

* You have provided a new quantitative analysis of cell viability in different oxygen concentrations by using Calcein AM/Propidium Iodide to stain live and dead cells. Data were plotted with arbitrary luminescence on the Y axis, and you have claimed that higher oxygen concentrations lead to a higher cell viability. However, there might be another explanation, i.e. the fact that aggregates cultured in high oxygen concentrations are larger (Figure 2D) may be the cause of the increased luminescence, suggesting that cell proliferation rather than absolute viability may be more influences by oxygen.

* The effective differentiation potential in derivatives from the three layers should have been tested. There is very little logic to test the "three lineage gene expression vs oxygen" if we do not know the actual capacity of forming unipotent derivatives of the three layers under directed differentiation. It is also not clear how these settings should be considered as representative of the in vivo development. Again, I reiterate my questions: what is the message that the readers should get from this paper? Should we start culture iPSCs in low oxygen concentation, due to the enhanced expression of differentiation markers? Should we keep culture them in high glucose concentration due to the higher growth rate and size of the aggregates? Which applications you envise from these different conditions? I understand that you are not willing to perform additional experiments, so here a written reply could at least be enough.

Author Response

Dear Reviewer,

We would like to thank the reviewer for insightful suggestions. Please find the attachment herewith.
